# Disentangling the Heterosis in Biomass Production and Radiation Use Efficiency in Maize: A Phytomer-Based 3D Modelling Approach

**DOI:** 10.3390/plants12061229

**Published:** 2023-03-08

**Authors:** Xiang Liu, Shenghao Gu, Weiliang Wen, Xianju Lu, Yu Jin, Yongjiang Zhang, Xinyu Guo

**Affiliations:** 1State Key Laboratory of North China Crop Improvement and Regulation, Key Laboratory of Crop Growth Regulation of Hebei Province, College of Agronomy, Hebei Agricultural University, Baoding 071000, China; 2Beijing Key Lab of Digital Plant, National Engineering Research Center for Information Technology in Agriculture, Beijing 100097, China; 3Information Technology Research Center, Beijing Academy of Agriculture and Forestry Sciences, Beijing 100097, China

**Keywords:** *Zea mays* L., grain yield, canopy structure, light interception, photosynthesis

## Abstract

Maize (*Zea mays* L.) benefits from heterosis in-yield formation and photosynthetic efficiency through optimizing canopy structure and improving leaf photosynthesis. However, the role of canopy structure and photosynthetic capacity in determining heterosis in biomass production and radiation use efficiency has not been separately clarified. We developed a quantitative framework based on a phytomer-based three-dimensional canopy photosynthesis model and simulated light capture and canopy photosynthetic production in scenarios with and without heterosis in either canopy structure or leaf photosynthetic capacity. The accumulated above-ground biomass of Jingnongke728 was 39% and 31% higher than its male parent, Jing2416, and female parent, JingMC01, while accumulated photosynthetically active radiation was 23% and 14% higher, correspondingly, leading to an increase of 13% and 17% in radiation use efficiency. The increasing post-silking radiation use efficiency was mainly attributed to leaf photosynthetic improvement, while the dominant contributing factor differs for male and female parents for heterosis in post-silking yield formation. This quantitative framework illustrates the potential to identify the key traits related to yield and radiation use efficiency and helps breeders to make selections for higher yield and photosynthetic efficiency.

## 1. Introduction

The increase in yield per unit ground area plays a crucial role in ensuring food security in the context of a sharp increase in global population and a gradual decrease in total arable land area. Heterosis, the superior performance of F1 hybrid progeny relative to parental phenotypes, is exhibited notably in maize for a wide range of traits [1], including leaf area [2], photosynthetic capacity [3,4], and yield [5]. The heterosis in grain yield is the synthetic result of interaction between plant morphological and physiological traits influenced by the environment over the whole growth season. Grain yield can be calculated as the product of accumulated photosynthetically active radiation (PAR) intercepted by the canopy over the grain-filling period and the radiation use efficiency (RUE) from the view of source [6], as well as to that of kernel weight, kernel number per ear, and ear density at harvest from the view of sink. Previous studies mainly focused on the heterosis of photosynthetic traits at the leaf level, leaf area at the plant level, and yield components. However, the heterosis in RUE has rarely been adequately quantified [7,8]. Incomplete information, as well as lack of precise and insufficient methods in determining and analyzing traits in traditional field experimental study, may impede our in-depth understanding of the relationship between crop structure and PAR interception, which in turn affects the precise quantification of RUE [9,10]. For example, Curin et al. [11] adopted the approach of Monteith [12] to analyze the heterosis of RUE and its causes, but they did not take the plant architecture into consideration when calculating PAR interception, thereby overlooking the difference in PAR interception and simply attributing the heterosis to the increase in grain yield.

Crop RUE, the efficiency of converting intercepted PAR to photo-assimilates of crop canopies, is largely dependent on plant architecture and canopy photosynthesis [6]. Optimal plant architecture [13] and increased leaf photosynthetic capacity [14] increase canopy photosynthetic efficiency, resulting in an improvement in crop yield per unit land area. The crop canopy photosynthesis model has been proven to be a powerful tool to quantify canopy photosynthetic capacity and simulate yield potential [14]. The Beer-Lambert rule has been mainly developed and applied to estimate light distribution to compute canopy photosynthesis by integrating leaf photosynthesis over the whole canopy and to calculate accumulated PAR interception by canopy to quantify RUE by obtaining the slope of linear regression against dry mass over a growth period. For example, Stewart et al. [15] evaluated the canopy light interception and photosynthesis for cultivars with contrasting plant architecture. Huang et al. [16] developed a light interception model and characterized the effect of plant architecture shaped by growth regulators on canopy PAR interception for a modern maize hybrid. The Beer-Lambert rule was further utilized to characterize the vertical distribution of light in maize canopy [17]. However, the simplification of the relationship between light distribution and canopy structure to exponential attenuation using LAI and the extinction coefficient limited the ability to accurately simulate the highly heterogeneous PAR distribution within the canopy. A simplified canopy photosynthesis model, without considering shading effects and plastic responses, was found prone to overestimate the intercepted PAR intensity, resulting in a further uncertainty in quantifying RUE [18]. Therefore, when investigating heterosis in radiation capture and photosynthetic production, detailed structural traits are urgently required to be considered in the canopy photosynthesis model.

With the wide application of computer graphics and three-dimensional visualization techniques in agricultural science in recent years, high-precision plant morphological modeling has gradually become possible, and its integration with the radiation model and leaf photosynthesis model can be used to quantify the contribution of morphological changes to RUE superiority. Rapid, accurate, and nondestructive acquisition of plant three-dimensional structures has become achievable using instruments and equipment, such as three-dimensional digitizers, multi-view visible light images, and LIDAR [19]. The three-dimensional crop canopy model not only accurately describes the geometric and morphological information at the organ scale, but it also fully considers the topology and spatial distribution of organs. Coupling crop three-dimensional canopy models with canopy light interception models has proven to be an effective method for assessing light interception and canopy photosynthetic capacity of plants with different structures [20,21,22], and it has been widely used in crops, such as rice [23], sorghum [24], sweet pepper [25], and maize [26]. A plant can be regarded as a set of interconnected phytomers, each of which consists of a leaf, a sheath, a node, and an internode [22]. The three-dimensional phytomer contains spatial coordinates and morphological parameters in three-dimensional space, all of which, at the plant scale, form a collection of phenotypic traits, topological structure, and spatial posture [27], thus providing a more accurate and detailed quantitative description of maize canopy structure. Phytomer-based three-dimensional modelling has been proven feasible to disentangle the contribution of phenotypic plasticity to light capture in maize–wheat intercropping systems [28], of bent shoots to photosynthetic production in individual rose [29], and of plant traits to light partitioning in maize/soybean intercropping system [30].

The objectives of this study were: (1) to develop a three-dimensional-phytomer-based modelling method for quantitatively characterizing the heterosis in morphological traits, light capture, and canopy photosynthetic production; and (2) to apply this method to disentangle the effects of canopy structure and leaf photosynthesis on above-ground biomass accumulation and RUE.

## 2. Materials and Methods

### 2.1. Experimental Setup and Environmental Condition

The field experiment was conducted in 2021 and 2022 at the experimental field of the Beijing Academy of Agriculture and Forestry Sciences in Haidian district, Beijing, China (39°56′ N, 116°16′ E). A randomized complete block design was used in this study. The maize hybrid Jingnongke 728 (JNK728) and its parental inbreds Jing 2416 (J2416) and JingMC01 (JMC01) were selected as the treatment, each of which has three replicates. The plant density is 6 plants m^−2^, having an equal row distance of 0.6 m. Each plot has a length and width of 6 m and 4.5 m, respectively. The seeds were sown on 26 June 2021 and 15 June 2022. The soil in the experimental field is brown sandy loam with total organic carbon 1.58%, total soil nitrogen of 1.34 g kg^−1^, Olsen phosphorus of 0.038 g kg^−1^, and available potassium of 0.091 g kg^−1^. The soil pH is 7.6, measured in H_2_O. The other field managements were conducted according to the local high-yield field. The number of appeared leaves were recorded for five consecutive plants of each plot every two days during the growing season. The silking and mature dates were 16 August and 14 October in 2021, 6 August and 4 October in 2022 for JNK728, 19 August and 17 October in 2021 for the third variant, 14 August and 12 October in 2022 for J2416, 19 August and 17 October in 2021 for the fifth variant, and 12 August and 10 October in 2022 for JMC01. During the maize growth season, average daily air temperature was 22.2 °C in 2021 and 23.4 °C in 2022, rainfall was 581.6 mm in 2021 and 405.7 mm in 2022, and total sunshine hours were 555.4 h in 2021 and 839.2 h in 2022 (Figure 1).

### 2.2. 3D Digitalization and Canopy Reconstruction by Assembling Phytomers

Three representative plants were selected for sampling in each plot every two days, starting from seedling to silking stage in 2021 and 2022. The north direction was recorded on each selected plant, then they were dug up and fixed it in a pot with wet soil fully filled to keep the plants fresh. The FastScan (Polhemus, Colchester, VT, USA) was used to obtain three-dimensional digitalized data for each maize plant. The phenotypic traits were first extracted following the approach by Wen et al. [31], and then each phytomer template, including the traits of organ in this phytomer and the spatial coordinates of each point, were added into the template database [32]. The t-distribution function was used to select the most similar phytomer template from the database according to the plant traits input, and then all the three-dimensional phytomers were assembled for three-dimensional canopy reconstruction. To reproduce the canopy environment in field and minimize the border effects on light interception, a three-dimensional canopy of 150 plants (10 × 15) was reconstructed for each measurement to simulate the PAR distribution.

### 2.3. Morphological Parameters for an Individual Leaf in Maize

Morphological parameters of individual leaves were extracted based on phytomers (Figure 2). Leaf length (*LL_i_*) is equal to the length of leaf midrib at the phytomer *i*. The leaf growth height (*HLBase_i_*), leaf top height (*HLTop_i_*), and leaf tip height (*HLTip_i_*) are the vertical distances from the leaf collar, leaf top, and leaf tip at the phytomer *i* to the ground, respectively. The height difference between leaf base and top (*HDBTop_i_*) was calculated as the difference between the *HLTop_i_* and *HLBase_i_*. The height difference between leaf base and tip (*HDBTip_i_*) was calculated as the difference between the *HLTip_i_* and *HLBase_i_*. The leaf tip level length (*Ltip_i_*) is equal to the projection length of total leaf midrib at the phytomer *i*. The leaf top level length (*Ltop_i_*) is equal to the projection length of leaf midrib from the collar to the highest point at the phytomer *i*.

### 2.4. Measurement and Parameterization of Photosynthetic Light Response Curve

Leaf photosynthetic light response curve (A-Q curve) was measured for the ear leaf and leaves at the third leaf above and below the ear leaf on clear days between 21 August and 25 August 2021 to represent the photosynthetic capacity of the middle, upper, and lower canopy for each plot without considering the heterogeneity within each layer. The PAR intensity was set to 2000, 1800, 1500, 1200, 1000, 750, 500, 250, 150, 100, 50, and 0 µmol m^−2^ s^−1^. The CO_2_ concentration in the leaf chamber was set to 400 µmol mol^−1^. After fitting to measurements, we estimated the net photosynthesis rate of leaf at the *i-*th phytomer of plant *j* at the time of *t* (*A_j,i_*_,*t*_, µmol m^−2^ s^−1^) using Ye’s model [33], as follows:(1)$$A_{j,i,t}=\alpha_{i}\frac{1-\beta_{i}PPFD_{j,i,t}}{1+\gamma_{i}PPFD_{j,i,t}}PPFD_{j,i,t}-R_{d,i}$$
where *PPFD_j,i_*_,*t*_ (µmol m^−2^ s^−1^) is the incident photosynthetic active radiation intensity for the leaf at the *i-*th phytomer at the time of *t*, *α_i_* is the initial slope of the A-Q curve, *β_i_* and *γ_i_* are non-dimensional parameters reflecting photoinhibition and light saturation, and *R_d,i_* is the dark respiration rate (µmol m^−2^ s^−1^). The maximum net photosynthesis rate (*A_max_*_,*i*_, µmol m^−2^ s^−1^) was further calculated, as below:(2)$$A_{max,i}=\alpha_{i}{\left(\frac{\sqrt{\beta_{i}+\gamma_{i}}-\sqrt{\beta_{i}}}{\gamma_{i}}\right)}^{2}-R_{d,i}$$

### 2.5. The PAR Distribution and Sunlit Leaf Area in the Three-dimensional Maize Canopy

The PAR distribution in the three-dimensional maize canopy was computed following the procedure developed by Gu et al. [34]. Direct and diffuse PAR interception was separately calculated for each facet of leaves at an hourly step. The area of sunlit leaves (*S*_sunlit,*i*_) was calculated by summing up the area of triangles (small facet as a basic unit involving in radiation computation), in which the angle between its normal vector and the direction of direct rays is higher than 90° [34]. More details of this model and software can be found in Gu et al. [34]. Based on this procedure, we added a virtual wall with a height equal to 60% of plant height surrounding the canopy to prevent the light penetration from the side as the real maize canopy in field. The output from the radiation module consists of day of year (DOY), plant number (*j*), leaf rank (*i*), hour (*t*), leaf area (*LA_j,i_*, cm^2^), intercepted photosynthetic photons by the *i*-th leaf during the hour of *t* (*IPP_j,i,t_*, µmol leaf^−1^ h^−1^), and sunlit leaf area (*LA*_sun,*j,i*_). The PAR distribution of maize canopy during the sampling interval of 2 days was computed using the three-dimensional canopy of previous measurement. To maximize the border row effects, the focal canopy consisting of nine plants (3 × 3) was selected to model canopy photosynthetic production.

### 2.6. Calculation of Above-Ground Biomass, Accumulated PAR Interception, and RUE

The output of the three-dimensional radiation module was used to calculate canopy photosynthesis rate (*A*_canDAY,*d*_, µmol m^−2^ s^−1^), daily above-ground biomass of maize canopy (*DM_d_*, g m^−2^ d^−1^), daily PAR interception by the canopy (*IPAR_d_*, MJ m^−2^ d^−1^), accumulated above-ground biomass (*ADM_d_*, g m^−2^), accumulated PAR interception (*AIPAR_d_*, MJ m^−2^), and RUE (g MJ^−1^) sequentially. The incident photosynthetic photon flux density of the leaf at phytomer *i* on plant *j* (*PPFD_j,i_*_,*t*_, µmol m^−2^ s^−1^) was calculated as follows:(3)$$PPFD_{j,i,t}=\frac{IPP_{j,i,t}}{3600LA_{j,i}10^{-4}}$$

The *A_j,i,t_* was integrated over each individual leaf and plant to obtain the instantaneous canopy photosynthesis rate (*A*_can,*t*_) using the following equation:(4)$$A_{\mathrm{can},t}=\frac{\sum_{j}^{M}\sum_{i}^{N_{j}}A_{j,i,t}LA_{j,i}}{M}PD$$
where *N_j_* is the number of leaves for the individual plant *j*, *M* is the total number of focal plants with a value of 9, and PD is the plant density with a value of 6 plants m^−2^.

The *A*_canDAY,*d*_ was calculated by integrating *A*_can,*t*_ over the daytime as follows:(5)$$A_{\mathrm{canDAY},d}=\overset{sunset}{\underset{t=sunrise}{\sum }}3600A_{\mathrm{can},t}$$

The *DM_d_* was calculated by:(6)$$DM_{d}=44A_{\mathrm{canDAY},d}C_{r}$$
where 44 is mole mass of CO_2_, and *C_r_* is the efficiency of converting CO_2_ to dry mass (0.41 for maize from the results by [35]).

The *IPAR_d_* was calculated as following equations:(7)$$IPAR_{j,i,d}=\frac{\sum_{t=sunrise}^{sunset}IPP_{j,i,t}}{4.55}10^{-6}$$
(8)$$IPAR_{j,d}=\overset{N_{j}}{\underset{i=1}{\sum }}IPAR_{j,i,d}$$
(9)$$IPAR_{d}=\frac{\sum_{j=1}^{M}IPAR_{j,d}}{M}PD$$
where *IPAR_j,i,d_* (MJ leaf^−1^ d^−1^) is the daily PAR interception by the leaf at the phytomer *i* of plant *j* in the center canopy, and *IPAR_j,d_* (MJ plant^−1^ d^−1^) is the daily PAR interception by the plant *j* in the center canopy, and the conversion factor of 4.55 µmol J^−1^ was used to convert PAR in µmol m^−2^ s^−1^ to PAR in J m^−2^ s^−1^ over the canopy [36]. The accumulated PAR interception by an individual leaf at the phytomer *i* of plant *j* over the entire growth season (*IPAR_j,i_*, MJ leaf^−1^) was calculated as the sum of *IPAR_j,i,d_*. The *ADM_d_* and *AIPAR_d_* are equal to the cumulative sum of *DM_d_* and *IPAR_d_* during the post-silking stage. The RUE was estimated as the quotient between *ADM_d_* and *AIPAR_d_*. The yield was assumed to be the accumulated above-ground biomass during the post-silking stage for all cultivars. For convenience of computation and comparison, the post-silking duration was set to be constant at 60 days.

The reconstructed canopy was divided into middle, upper, and lower canopy, in which the middle includes the ear leaf and the first leaf above and below the ear leaf, the upper includes leaves above the middle canopy, and the lower includes leaves below the middle canopy (Figure 3). The total intercepted PAR by the upper (*IPAR*_upp_, MJ m^−2^), middle (*IPAR*_mid_, MJ m^−2^), lower (*IPAR*_low_, MJ m^−2^), and entire canopy (*IPAR*_can_, MJ m^−2^) over the growth season were calculated by upscaling the sum of *IPAR_j,i_* of corresponding leaves to the canopy level.

### 2.7. Contribution of Canopy Structure and Photosynthetic Traits to Heterosis in Yield and RUE

To quantify the contribution of canopy structure and leaf photosynthetic capacity to heterosis in yield and RUE, we simulated light capture and photosynthetic production for different scenarios by formulating different combinations of three-dimensional canopy structure and photosynthetic traits (Table 1).

The contribution of canopy structure and photosynthetic capacity were separately calculated as the relative change in simulated yield and RUE, resulting from resetting either the canopy structure or photosynthetic capacity of JNK728 to its parental inbreds. The relative change in percentage caused by canopy structure and leaf photosynthesis compared to J2416 (*C_c_*_,*J*2416_, *C_p_*_,*J*2416_) and JMC01 (*C_c_*_,*JMC*01_, *C_p_*_,*JMC*01_) were expressed as the following equations:(10)$$C_{c,J2416}=\frac{Y_{JNK728}-Y_{A1}}{Y_{JNK728}-Y_{J2416}}100$$
(11)$$C_{p,J2416}=\frac{Y_{JNK728}-Y_{B1}}{Y_{JNK728}-Y_{J2416}}100$$
(12)$$C_{c,JMC01}=\frac{Y_{JNK728}-Y_{A2}}{Y_{JNK728}-Y_{J\mathrm{MC}01}}100$$
(13)$$C_{p,JMC01}=\frac{Y_{JNK728}-Y_{B2}}{Y_{JNK728}-Y_{J\mathrm{MC}01}}100$$
where *Y_JNK_*_728_, *Y_J_*_2416_, *Y_JMC_*_01_, *Y_A_*_1_, *Y_A_*_2_, *Y_B_*_1_, and *Y_B_*_2_ indicate the simulated yield (Mg ha^−1^) or RUE (g MJ^−1^) in different scenarios.

### 2.8. Statistical Analysis

The canopy photosynthesis model was developed using the R language. ANOVA analysis and multiple comparisons were performed using SPSS 26 statistics software (IBM, Armonk, NY, USA). Treatment means were separated using least significant differences (LSD) at the 5% level of significance.

## 3. Results

### 3.1. Morphological Traits

The plant architecture of JNK728 was found to be considerably different with J2416 and JMC01 (Figure 3). The results showed that plant height and leaf area between cultivars are significantly different, and the values are in descending order of JNK728 > JMC01 > J2416. The ear height and photosynthetic leaf area of JNK728 were found to be significantly higher than those of its parental inbreds, with an average increase of 51.15% and 43.74%, respectively. There was no significant difference between cultivars in leaf inclination angle (Table 2).

The *LL_i_*, *HDBTop_i_*, *HDBTip_i_*, *Ltip_i_*, and *Ltop_i_* showed an unimodal distribution against the phytomer rank. The *LL_i_* of JNK728 were significantly higher than J2416 across phytomers and JMC01 below the phytomer rank of 15, above which the difference between JNK728 and JMC01 was not significant (Figure 4A). *HDBTop_i_* reached the maximum at phytomer 14 (67.68 cm, 60.43 cm, and 74.96 cm) and then gradually decreased for JNK728, J2416, and JMC01. *HDBTop_i_* of JNK728 was found to be significantly higher than J2416 for the leaves in the canopy top with a phytomer number higher than 15 (Figure 4B). A similar trend was also observed for *HDBTip_i_* (Figure 4C). The *Ltop_i_* reached the maximum at the phytomer 13 for JNK728 (47.54 cm) and J2416 (38.18 cm) and at the phytomer 12 for JMC01 (61.67 cm). The *Ltop_i_* for the ear leaf in JNK728 was significantly higher than that in J2416 and JMC01 (Figure 4D). *Ltip_i_* peaked at phytomer 11 (73.74 cm, 54.25 cm, and 74.04 cm) and then decreased for JNK728, J2416, and JMC01. The JNK728 had a significantly higher *Ltip_i_* than its parental inbreds for most phytomers (Figure 4E).

### 3.2. Sunlit Leaf Area and Incident Photosynthetic Active Radiation Intensity

The *S*_sunlit,*i*_ increased from the bottom phytomer to one to three phytomers above the ear with a maximum of 333.45–430.21 cm^2^, and then it decreased in relation to the top phytomer (Figure 5A,C,E). The JNK728 was found to have a substantially higher *S*_sunlit,*i*_ than J2416 and JMC01 in spite of some fluctuations, especially for phytomers above the ear at 12:00 and 16:00 (Figure 5C,E). *S*_sunlit,*i*_ of JNK728 was found to be 92% and 72% higher than that of J2416 and JMC01 on average. The PPFD showed an increasing pattern against increasing phytomer ranks (Figure 5B,D,F). Despite an overall decrease, the leaves of JNK728 were found to have a significantly higher PPFD at ear phytomer (77% and 105%) and at three phytomers (2% and 29% on average) above the ear at 12:00 than at J2416 and JMC01 (Figure 5D).

### 3.3. PAR Interception over Growth Season

*IPAR_d_* increased with the plant growth and reached the maximum at the onset of silking stage in 2021 and 2022. *IPAR_d_* over the entire growth season in 2021 and 2022 was in descending order: JNK728 > JMC01 > J2416, suggesting that the canopy of JNK728 intercepted more PAR than JMC01 and J2416. *IPAR_d_* showed substantial fluctuations under different solar conditions. The difference between cultivars became lager when *IPAR_d_* was higher. The *IPAR_d_* of JNK728 was increased by 23% and 35% in 2021, and by 51% and 73% in 2022, on average, in comparison with JMC01 and J2416, respectively (Figure 6).

*IPAR_j,i_* against phytomer ranks followed a distinctly bell-shaped pattern. Despite the slight difference between the leaves under the ear phytomer for different cultivars, *IPAR_j,i_* of JNK728 above the ear leaf was found to be significantly higher than J2416 and JMC01 in both 2021 and 2022 (Figure 7).

The effects of cultivars, canopy layers, and their interaction on total intercepted PAR were significant (*p* < 0.05) in 2021 and 2022. *IPAR*_can_ of JNK728 was increased by 20–27% and 10–15% in comparison with J2416 and JMC01 across two years. The different canopy layers in JNK728 intercepted significantly more PAR than J2416, whereas the increase was partly offset by the decrease in *IPAR*_mid_ in 2021 and of *IPAR*_upp_ compared to JMC01 (Table 3).

### 3.4. Leaf Photosynthetic Capacity

The simulations of net photosynthesis rate by Ye’s model were in good agreement with observations across cultivars and phytomer ranks (R^2^ > 0.99, Figure 8). Cultivar and phytomer rank significantly affected the α, *A*_max_, and *R*_d_ of leaves in 2021 (*p* < 0.05) (Table 4). There was no interaction between cultivar and phytomer rank. The SPAD was only significantly affected by the cultivars (*p* < 0.05). The upper leaf of JNK728 was found to have a significantly higher α than J2416 and JMC01, with an increase of 16% and 36%, suggesting a higher photosynthetic efficiency under low irradiance. *A*_max_ of JNK728 was significantly higher than J2416 and JMC01 across canopy layers, with an overall increase of 9% and 12%. JNK728 had a significantly higher *R*_d_ than JMC01 in the upper leaf, and this was also significantly higher than J2416 and JMC01 in the ear leaf. The SPAD of JNK728 was significantly higher than J2416 and JMC01 across canopy layers (Table 4).

### 3.5. Contribution of Canopy Structure and Photosynthetic Capacity to Heterosis in Yield and Radiation Use Efficiency

The simulated yield of JNK728 (9.96 Mg ha^−1^) was significantly higher than J2416 (7.37 Mg ha^−1^) and JMC01 (7.83 Mg ha^−1^) (Figure 9A,C). The canopy structure and photosynthetic capacity contributed 70% (i.e., (9.96 − 8.15)/(9.96 − 7.37) × 100%) and 36% (i.e., (9.96 − 9.03)/(9.96 − 7.37) × 100%) to the yield increase compared to J2416, as well as 42% (i.e., (9.96 − 9.07)/(9.96 − 7.83) × 100%) and 74% (i.e., (9.96 − 8.38)/(9.96 − 7.83) × 100%) compared to JMC01. The JNK728 was found to have a higher RUE, with an increase of 13% and 17% in comparison with J2416 and JMC01 (Figure 9B,D). The canopy structure and photosynthetic capacity contributed 21% (i.e., (4.07 − 3.97)/(4.07 − 3.60) × 100%) and 87% (i.e., (4.07 − 3.66)/(4.07 − 3.60) × 100%) to the RUE increase compared to J2416, and 5% (i.e., (4.07 − 4.04)/(4.07 − 3.49) × 100%), as well as 116% (i.e., (4.07 − 3.40)/(4.07 − 3.49) × 100%) compared to JMC01.

## 4. Discussion

### 4.1. The Heterosis in Structural and Photosynthetic Traits at Different Levels

The present study revealed that heterosis occurred in leaf size (Figure 4), total leaf area (Table 2), photosynthetic capacity at the leaf level (Table 4), and photosynthetic efficiency at the canopy level (Figure 9). The heterosis was also confirmed by Tollenaar et al. [37] for leaf size and total leaf area, Wang et al. [38] for leaf photosynthesis, and Munaro et al. [39] for RUE. The heterosis for individual leaf area in maize was found to be resultant from the complementation of dominance effects on cell size and overdominance for cell number [40] and further from the effects of *CNR* genes on leaf epidermal cell number [41]. Despite disagreement with results on leaf photosynthesis reported by Ahmadzadeh et al. [42] in the silking stage, the heterosis became apparent two weeks after silking, and it was increasingly larger toward maturity. This might be associated with increased expression levels of carbon fixation genes, thus promoting carbon fixation and biomass accumulation [43]. The maize metabolic and proteomic data further confirmed the key role of enzymes involved in photosynthetic pathways in improving carbon assimilation [44].

Improved RUE via optimizing canopy structure is a major aspect of improving yield potential [45,46]. The superiority of hybrids in RUE could be attributed to the higher magnitude of the relative increase in photosynthetic production than that in PAR interception during a certain period. In the present study, the accumulated above-ground biomass of JNK728 was increased by 39% and 31% compared to J2416 and JMC01, while accumulated PAR was increased by 23% and 14%, correspondingly (Figure 9). This is in agreement with the previous field experimental results that higher yielding maize lines have a higher RUE [46,47]. Liu et al. [46] attributed the larger RUE to the improved canopy structure with optimized maize plant type at high plant density. The morphological improvements were expressed in terms of higher leaf area [46], more erect stature [47], and optimized leaf architecture [48]. This evidence supports our results on modifications of leaf area and leaf architecture (Figure 4 and Table 2), but not on leaf angle (Table 2). In the present study, the JNK728 with significantly higher leaf area over phytomer ranks could receive higher light intensity and intercept more light than its parental inbred lines (Figure 7). In addition, the larger space above the ear may avoid mutual shading of leaves [49] and thus improve the light transmission [48]. This is in agreement with our results that the ample space occupied by each individual leaf (Figure 4B,C) avoids inter-plant competition for light and allows for more light penetration into the lower canopy (Figure 5). With these traits, PAR can be captured by leaves in the middle and lower canopy (Table 3), which are considered to be the major source of photo-assimilates for grain [16].

### 4.2. The Capability of Three-Dimensional-Phytomer-Based Canopy Photosynthesis Model in Simulating Light Capture and Photosynthetic Production

The three-dimensional-phytomer-based model has strong competence in characterizing phenotypic traits at the level of organ and in upscaling light capture and photosynthetic production from the individual leaf facet to the canopy in a fine resolution after coupling with the leaf photosynthesis model. Despite a slight difference in plant architecture between JNK728 and JMC01 (Table 2 and Figure 3), the significant difference in leaf architecture (Figure 4), sunlit leaf area (Figure 5), accumulated PAR interception (Figure 6 and Figure 7 and Table 3), biomass accumulation (Figure 9), and RUE (Figure 9) were captured by the three-dimensional-phytomer-based model. Our previous studies have demonstrated that the three-dimensional model is reliable in simulating vertical distribution of PAR in maize canopy across a large range of environments (e.g., plant densities, solar conditions, and experimental sites) and is capable of predicting the canopy photosynthetic production for different cultivars with similar canopy architecture [34,50]. The traditional models, based on the Beer-Lambert rule regard the canopy as a big leaf, calculate the accumulated PAR interception using the measured or calculated fraction of PAR; however, they could overestimate the PAR interception and overlook its difference between dense canopies [34]. Therefore, our three-dimensional-phytomer-based canopy photosynthesis model is reliable and powerful in predicting light interception and photosynthesis from leaf facet to canopy.

### 4.3. Implications for Breeding for Higher Yield and Photosynthetic Efficiency

Quantifying the role of canopy structure and leaf photosynthesis in driving heterotic effects helps to better understand the eco–physiological mechanisms underlying heterosis. Many studies analyzed the possible reasons for heterosis in grain yield qualitatively from the aspects of resource supply and capture, converting efficiency from resource to biomass, as well as biomass partitioning [37,39], but they did not quantify their contributions to heterosis in yield and RUE. However, traits from these aspects highly aggregate the complex interaction between canopy structure and leaf photosynthesis in three-dimensional space, necessitating distinguishing the separate contributions of canopy structure and photosynthetic capacity to the heterosis. The present study provides a methodology to quantify them. We found that the dominant contributing factors differ when comparing male and female parents on heterosis in yield, but not in RUE (Figure 9). For the male parent (i.e., J2416) with the lowest leaf area, but less photosynthetic capacity, heterosis in the canopy structure plays a dominant role in improving yield. For the female parent (i.e., JMC01) with the lowest photosynthetic capacity, but less leaf area, heterosis in leaf photosynthetic capacity contributes more than canopy structure. This quantitative framework could allow us to identify the key traits related to yield and RUE and help breeders make selections for higher yield and photosynthetic efficiency.

### 4.4. Potential Limitations

The current method moves a step forward in disentangling the source of heterosis in photosynthetic production and efficiency. However, there are still some aspects that need to be further considered. First, we assumed that the simulated yield was equal to the accumulated above-ground biomass during post-silking 60 days and did not consider the decline of photosynthesis during this period in this study. Leaf photosynthesis rate of both hybrids and their parental inbred lines declined during grain-filling stage, but their differences increased from silking to maturity [42]. The duration of the stay-green [37] and the grain-filling [51] of hybrids were also found to be higher than those of their parental inbreds. The absence of considering actual grain-filling duration and leaf longevity aggravated the discrepancy between simulated yield and actual yield, and, hence, it possibly underestimated the magnitude of heterosis in accumulated above-ground biomass and PAR. Second, although the contribution of canopy structure and leaf photosynthesis to yield and RUE has been separately quantified, it remains unclear what the structural traits are and how they matter in promoting light capture and yield formation. This provides a precise and practical breeding target for higher yield and photosynthetic efficiency, as designing maize ideotype through regulating leaf size and angle has already become feasible [13,52]. Therefore, the contribution of each specific trait to the heterosis needs to be quantified subsequently. Third, the heterosis in maize is also influenced by environmental factors, such as temperature [53], light [54], and soil water [55]. Responses of organ development, expansion, and photosynthesis to those environmental stimuli need to be incorporated into the current three-dimnsional-phyotmer-based maize model via developing the dynamically growing function.

## 5. Conclusions

A new method was developed by the present study to separate the effects of morphological changes and leaf photosynthetic improvement on heterosis in biomass production and RUE in maize. Using the three-dimensional-phytomer-based canopy photosynthesis model, we demonstrated that photosynthetic improvement plays a dominant role in increasing post-silking RUE and that the dominant contributing factor differs for male and female parents on heterosis in post-silking yield formation. Based on the structural traits in detail, we showed that the increases in leaf length, sunlit leaf area, and intercepted PAR intensity facilitate the heterosis in light capture by the canopy. To our knowledge, this is the first study to quantify the contribution of canopy structure and leaf photosynthetic capacity to heterosis in yield and RUE. Identifying key traits related to and quantifying their contributions to yield formation and photosynthetic efficiency will help us better understand the eco–physiological mechanism underlying heterosis.

## Figures and Tables

**Figure 1 plants-12-01229-f001:**
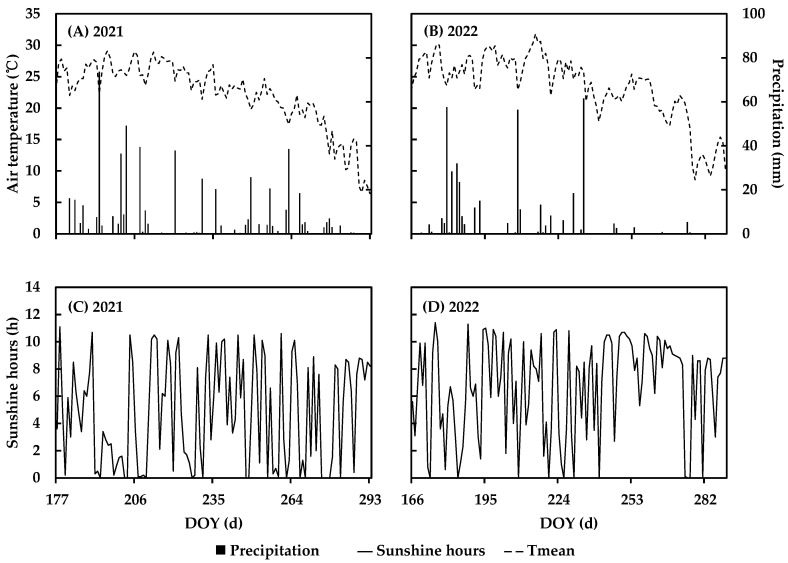
Daily mean air temperature, rainfall, and sunshine hours in maize growing season at Haidian in 2021 (**A**,**C**) and 2022 (**B**,**D**).

**Figure 2 plants-12-01229-f002:**
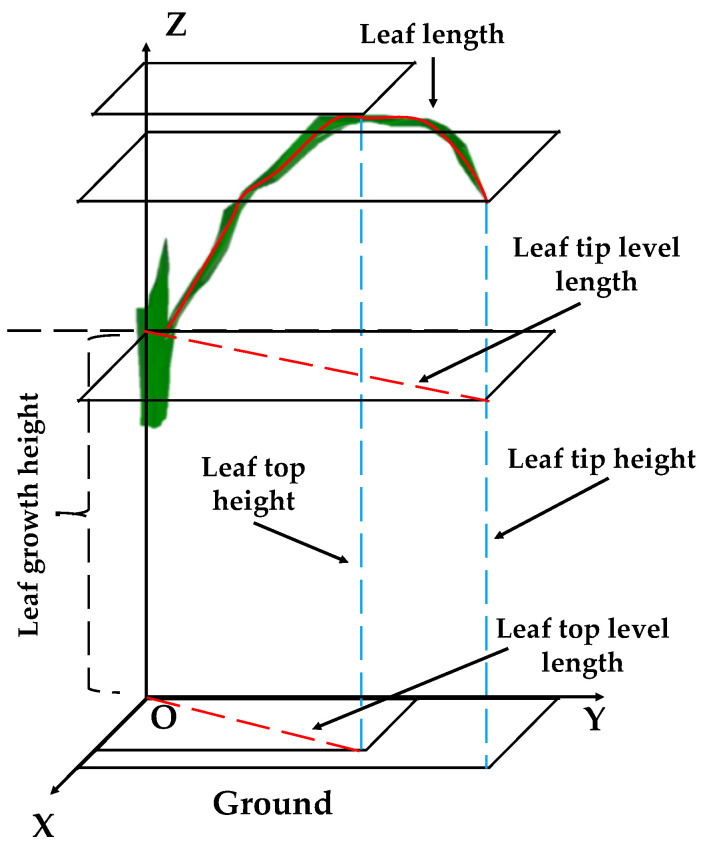
Schematic representation of the leaf length, leaf growth height, leaf top height, leaf tip height, leaf tip level length, and leaf top level length in a three-dimensional phytomer.

**Figure 3 plants-12-01229-f003:**
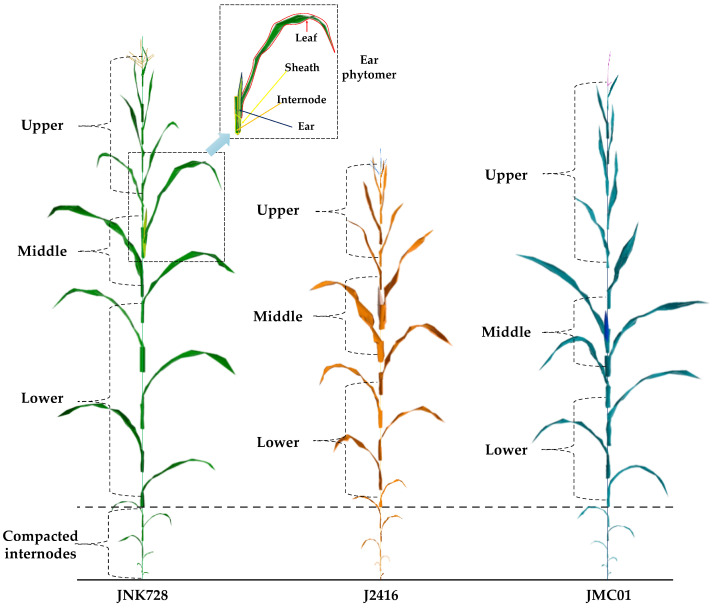
Visualization of three-dimensional architecture of maize plant for JNK728, J2416, and JMC01 on 26 August 2021 in the beginning of the post-silking stage.

**Figure 4 plants-12-01229-f004:**
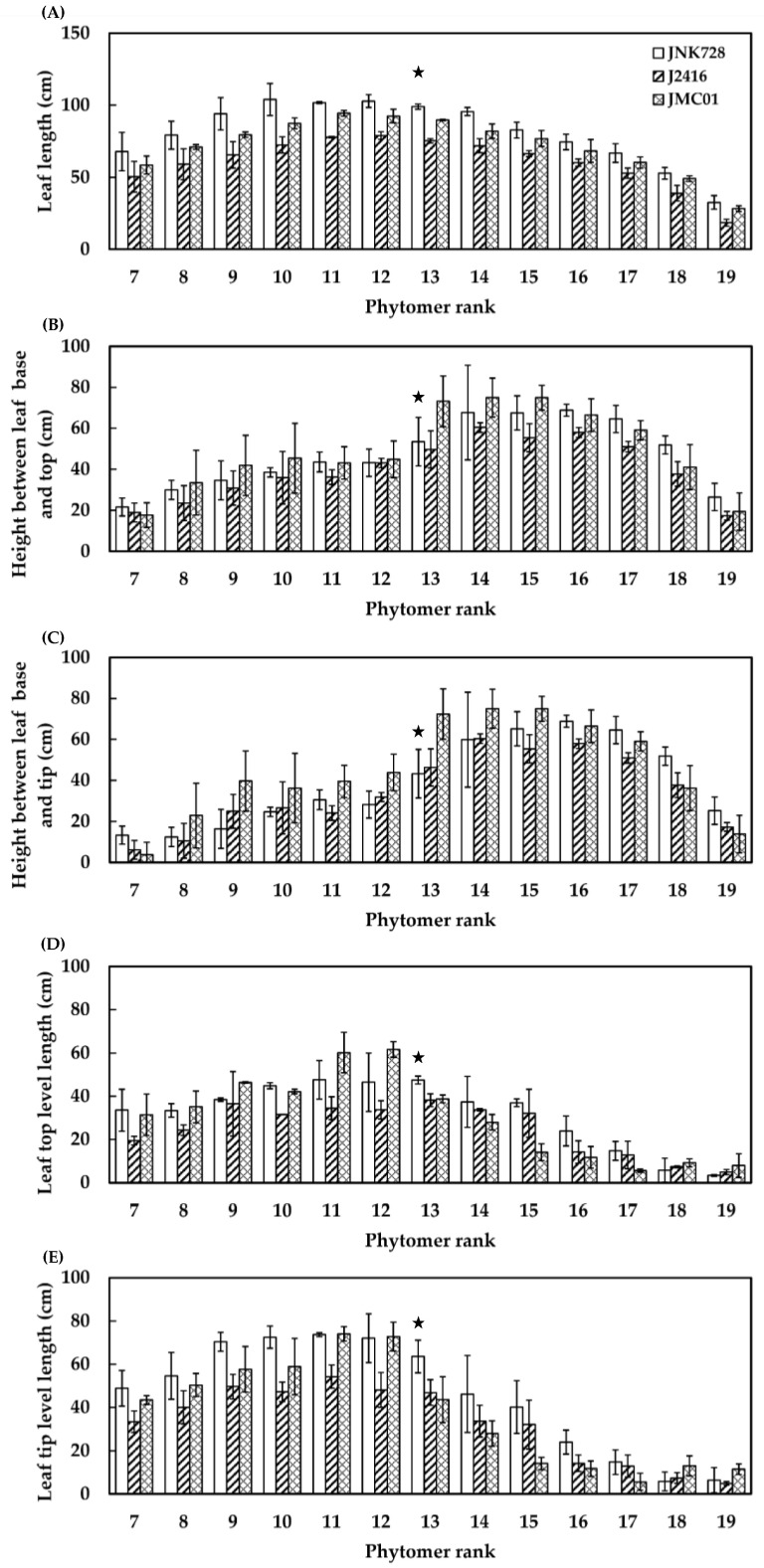
Leaf length (**A**), leaf base–top distance (**B**), leaf base–tip distance (**C**), leaf base–top length (**D**), and leaf base–tip length (**E**) at different phytomer ranks in 2021. Error bars indicate SE. The stars indicate ear positions.

**Figure 5 plants-12-01229-f005:**
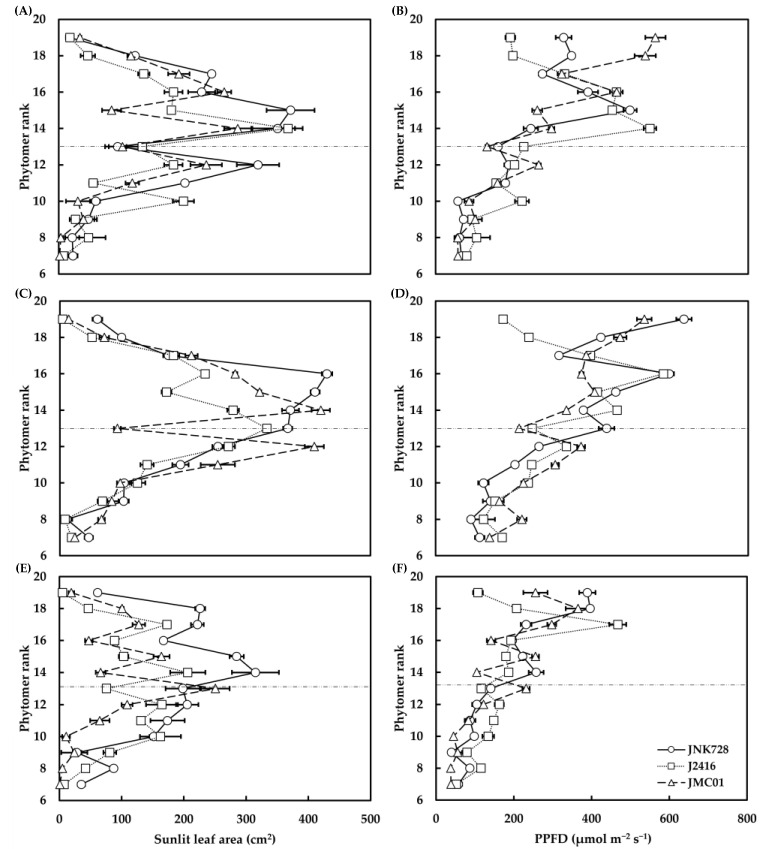
Sunlit leaf area (**A**,**C**,**E**) of and PAR interception (**B**,**D**,**F**) by individual leaves at different phytomers at 08:00 (**A**,**B**), 12:00 (**C**,**D**), and 16:00 (**E**,**F**) on 21 August 2021 for different cultivars during post-silking period. Error bars indicate the SE. Dashed lines indicate the position of the ear.

**Figure 6 plants-12-01229-f006:**
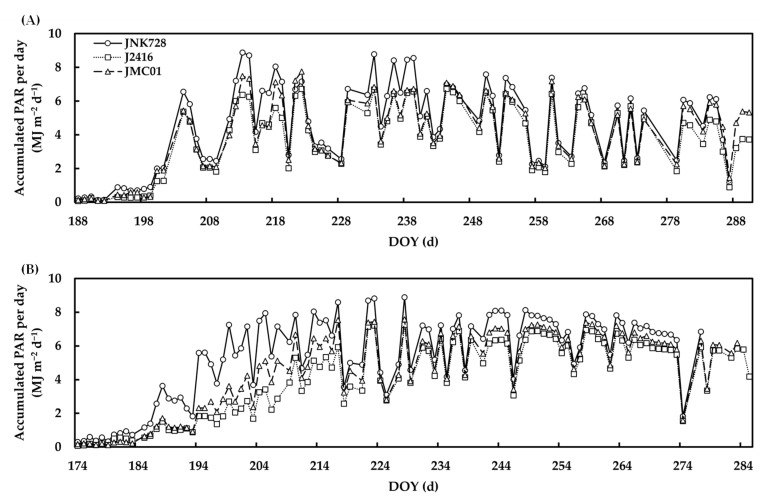
Accumulated PAR per day for JNK728 (open circles), J2416 (open squares), and JMC01 (open triangles) over the entire growth period in 2021 (**A**) and 2022 (**B**).

**Figure 7 plants-12-01229-f007:**
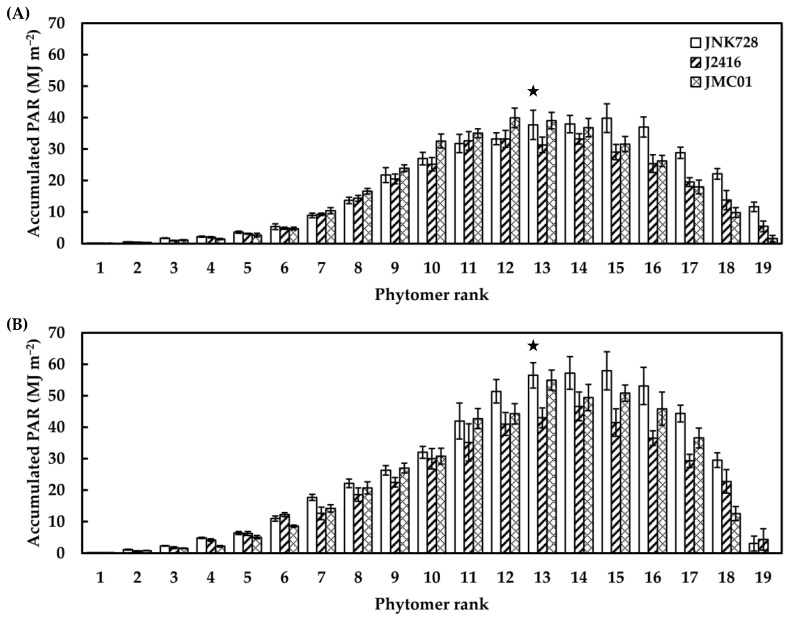
Accumulated PAR interception by individual leaves at different phytomers for JNK728, J2416, and JMC01 over the entire growth period in 2021 (**A**) and 2022 (**B**). Error bars indicate the SE. Stars indicate the position of the ear.

**Figure 8 plants-12-01229-f008:**
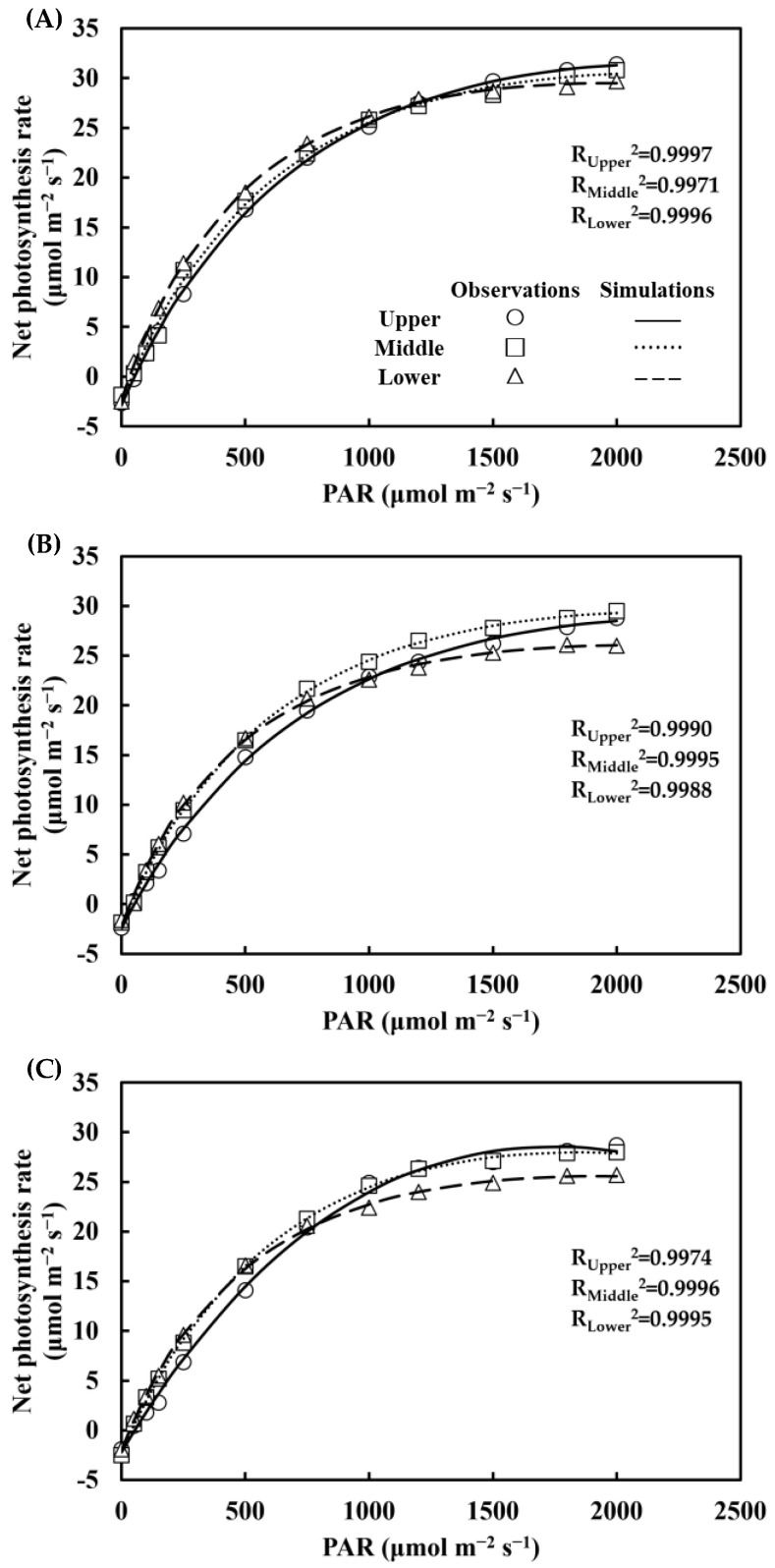
Observations (circles, squares, and triangles) and simulations (bold lines, dotted lines, and dashed lines) of net photosynthesis rate for the upper leaf, ear leaf, and lower leaf at different levels of PPFD for JNK728 (**A**), J2416 (**B**), and JMC01 (**C**) at the beginning of the silking stage in 2021.

**Figure 9 plants-12-01229-f009:**
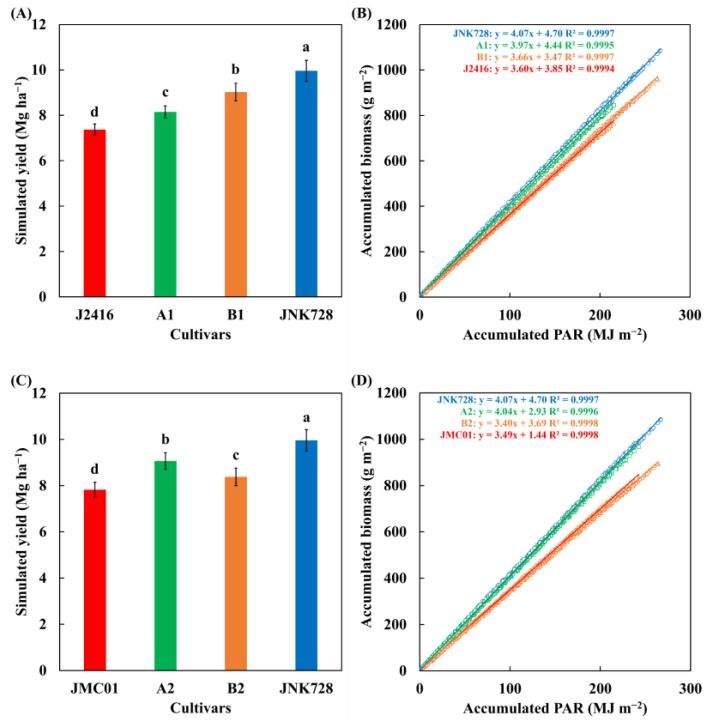
Simulated yield (**A**,**C**) and relationship (linear regression) between accumulated above-ground biomass and accumulated PAR interception (**B**,**D**) for different scenarios. Error bars indicate the SE. Different letters denote significant difference at *p* < 0.05.

**Table 1 plants-12-01229-t001:** Simulation scenarios for assessing the contribution of canopy structure and leaf photosynthetic capacity to heterosis in yield and RUE.

Scenarios	3D Canopy Structure	Leaf Photosynthetic Capacity
JNK728	JNK728	JNK728
J2416	J2416	J2416
JMC01	JMC01	JMC01
A1	J2416	JNK728
A2	JMC01	JNK728
B1	JNK728	J2416
B2	JNK728	JMC01

Note: A1 and A2 were simulated using the three-dimensional canopy structure of J2416 and JMC01 and the photosynthetic capacity of JNK728, while B1 and B2 were simulated using the photosynthetic capacity of J2416 and JMC01 and the three-dimensional canopy structure of JNK728.

**Table 2 plants-12-01229-t002:** Morphological parameters of maize plants for JNK728, J2416, and JMC01 in 2021.

Index	JNK728	J2416	JMC01
Plant height (cm)	266.37 ± 5.81 a	187.50 ± 3.43 c	251.68 ± 7.68 b
Phytomer number	19	19	19
Rank of ear	13	13	12
Ear height (cm)	106.85 ± 6.32 a	74.29 ± 0.69 b	67.42 ± 7.08 b
Ear length (cm)	34.57 ± 2.96 a	22.64 ± 2.24 b	33.42 ± 0.63 a
Leaf inclination angle (°)	66.7 ± 1.9 a	66.3 ± 1.6 a	65.3 ± 4.1 a
Leaf area (m^2^)	0.72 ± 0.06 a	0.46 ± 0.06 c	0.58 ± 0.04 b
Photosynthetic leaf area (m^2^)	0.23 ± 0.011 a	0.16 ± 0.008 b	0.16 ± 0.009 b

Note: Different letters denote significant difference at *p* < 0.05.

**Table 3 plants-12-01229-t003:** Total intercepted PAR by different canopy layers (P) for different maize cultivars (C).

Cultivars	Total Intercepted PAR (MJ m^−2^)							
	2021				2022			
	Upper	Middle	Lower	SUM	Upper	Middle	Lower	SUM
JNK728	139.60 aA	108.94 bC	116.70 aB	365.24 a	186.61 bA	165.08 aB	166.21 aB	517.90 a
J2416	93.25 cB	97.86 cB	113.51 aA	304.62 c	134.45 cB	130.57 cB	144.29 bA	409.32 c
JMC01	124.13 bA	114.10 aB	93.97 bC	332.20 b	195.33 aA	141.91 bB	111.32 cC	448.56 b
C	112.16 *			79.46 *	204.64 *			116.90 *
P	48.45 *			-	173.76 *			-
C × P	85.85 *			-	116.63 *			-

Note: Different letters denote significant difference at *p* < 0.05. Small letters indicate differences between different cultivars. Capital letters indicate differences between different layers. F values and significance levels (* *p* < 0.05) were listed in the bottom three rows.

**Table 4 plants-12-01229-t004:** Parameters of the photosynthetic light response curve of leaves at different phytomer ranks (P) for different maize cultivars (C) from 21 August to 25 August in 2021.

Cultivars	α			*A*_max_ (µmol m^−2^ s^−1^)			*R*_d_ (µmol m^−2^ s^−1^)			SPAD		
	Upper leaf	Ear Leaf	Lower Leaf	Upper Leaf	Ear Leaf	Lower Leaf	Upper Leaf	Ear Leaf	Lower Leaf	Upper Leaf	Ear Leaf	Lower Leaf
JNK728	0.057 aB	0.064 aB	0.074 aA	31.48 aA	30.54 aA	29.48 aB	2.88 aA	2.64 aA	2.34 aA	68.9 aA	67.9 aA	63.7 aB
J2416	0.049 bB	0.060 aA	0.068 aA	28.86 bA	29.43 bA	26.06 bB	2.45 aA	2.17 bA	2.24 aA	60.5 bA	57.3 bA	56.1 bA
JMC01	0.042 cC	0.057 aB	0.063 aA	28.54 bA	27.95 cA	25.59 bB	2.20 bA	2.24 bA	1.98 aA	52.1 cA	54.2 bA	54.5 bA
C	15.187 *			60.693 *			9.3 *			52.19 *		
P	46.508 *			45.948 *			4.156 *			1.701		
C × P	0.798			2.785			0.891			1.890		

Note: Different letters denote significant difference at *p* < 0.05. Small letters indicate differences between different cultivars. Capital letters indicate differences between different layers. F values and significance levels (* *p* < 0.05) were listed in the bottom three rows.

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
