# Peer review of "Disentangling the Heterosis in Biomass Production and Radiation Use Efficiency in Maize: A Phytomer-Based 3D Modelling Approach"

_plants, 2023, doi:10.3390/plants12061229_

Round 1
Reviewer 1 Report
Dear authors,
your article is generally well prepared. But I see a weak point in the Discussion section. This passage needs to be improved. Discuss your results with the professional literature in more detail. Some additional notes are in the attached pdf.
Best regards.

Reviewer 2 Report
The manuscript developed a quantitative framework based on a phytomer-based 3D canopy photosynthesis model to separate the effects of morphological changes and leaf photosynthetic improvement on heterosis in biomass production in Maize. It's an interesting work in plant phenomics. Some issues should be addressed before its publication.
1. The reason that the phytomer-based 3D model can disentangle the heterosis is suggested to be explained.
2. This study concluded that the increase of leaf length and sunlit leaf area facilitates the heterosis based on the analysis of traits from three plants JNK728, J2416 and jmc01. But we don't know whether other plants can also get the same conclusion. Why don't choose more plants to get a higher statistical result for the analysis?
3. As a tool development for plant phenomics, the related material and tool are suggested to be public for user on Github.
Round 2
Reviewer 1 Report
Lines 38 and 46 - the same abbreviation is explained.
I consider the article eligible for publication.
Author Response
Minor comments
Point 1: Lines 38 and 46 - the same abbreviation is explained.
Response 1: Thanks. We realized that this is not the first appearance of this abbreviation, so we removed the abbreviation and changed “canopy light interception” to “PAR interception”. See line 49.
Reviewer 2 Report
There is no further question for the manuscript.
Author Response
We have carefully checked the english language and style over the entire text and made modifcations accordingly. All revisions were made using the “Track changes” function.